# Neural Knowledge Base Repairs

Thomas Pellissier Tanon and Fabian Suchanek

Télécom Paris, Institut Polytechnique de Paris

**Abstract.** The curation of a knowledge base is a crucial but costly task. In this work, we suggest to make use of the advances in neural network research to improve the automated correction of constraint violations. Our method is a deep learning refinement of [23], and similarly uses the edits that solved some violations in the past to infer how to solve similar violations in the present. Our system makes use of the graph content, literal embeddings, and features extracted from Web pages to improve its performance. The experimental evaluation on Wikidata shows significant improvements over baselines.

## 1   Introduction

The past years have seen the rise of large knowledge bases, completed by a crowd of contributors like Wikidata [28] or Freebase [8], or automatically filled from extraction and conversion pipelines like Yago [27] or DBpedia [4]. Both kind of knowledge bases often contain errors, originating from edge cases in the conversion pipelines, good faith mistakes, or vandalism in the crowd-sourced content. Often, knowledge bases contain a constraint system in order to fight such problems, starting from the OWL distinction between object properties and datatypes properties, and including, e.g., domain and range constraints or more complex expressions such as cardinality constraints and conflict declarations. These constraints are often violated in practice. For example, Wikidata has 1M "domain" constraint violations and 4.4M "single value" constraint violations as of March 20th, 2020. Thus, there is a need for tools to help the knowledge base curators repair these violations in an automated or a semi-automated way.

Recent work [23] shows that, in the case of an actively curated knowledge base like Wikidata, it is possible to use the repairs that have been done in the past in order to learn the repairs for the current constraint violations. The work provides both a formalism and a first algorithm based on rule learning to this end. However, the experiments on Wikidata show that there is still room for improvement: The approach was not able to provide anything meaningful when RDF literals where involved, and the user study presents a low agreement score (less than 50%) for a lot of constraint types.

Hence, we explore in this work a new approach to learn how to repair constraint violations using the edit history, building upon [23]. Our method takes as input a KB with its edit history, a set of constraints, and the statements of the KB that constitute violations of the constraints. It produces as output suggestions of statements to add to the KB or to remove from the KB so that the

constraint violation disappears. Our new algorithm is based on deep learning, and brings two key advantages over [23]: First, our embeddings take into account the data of the KB in a holistic fashion, as opposed to being limited to the pieces of data captured by logical rules. Second, we are able to make use of data that would be hard to integrate into rule learning approaches, including, e.g., the textual content in the knowledge base. We also keep the ability from [23] to work at the scale of large knowledge bases. Our evaluation on Wikidata shows significant improvements against the state of the art. We also conduct detailed ablation studies to justify our architecture. Finally, we improve the Wikidata evaluation dataset that was introduced by [23], and release it in an easy to use format in the hope that it can be useful to evaluate such systems.

This paper is structured as follows: Section 2 discusses related work, and Section 3 introduces preliminaries. Section 4 presents a baseline. Section 5 explains our approach and Section 6 evaluates it, before Section 7 concludes.

## 2   Related work

Our work aims at repairing constraint violations in a collaboratively edited knowledge base. It builds on, and improves upon, the approach presented in [23]. Several other approaches are related to this endeavor.

**Knowledge base cleaning.** Several approaches have been developed to repair constraint violations in knowledge bases. Active integrity constraints [11] aim at providing a set of possible repair actions to each constraint. An application to description logic knowledge bases has been presented in [24]. These approaches assume that the user provides a set of possible corrections. In our work, in contrast, we want to learn these corrections automatically from the edit history. Again other works are interactive, and ask the user questions to quickly find a correction [6,7,3,2]. Our work, in contrast, learns the repair directly from the edit history.

Again other approaches use the data in the KB itself in order to improve it. [9] uses knowledge base embeddings, lexical distance, and constraint-based refinements to predict corrections. Our work goes beyond the state of the current KB and learns from the edit history instead.

[22] uses statistics to find errors in the knowledge base and to add new types. [18] exploits the graph structure to find wrong type relations. [21] cleans *sameAs* relations based on the shape of the existing identity and on differences in the graph. [17] uses graph and text distances to look for values for a given subject and predicate. [20] fixes the subject or object of existing relations based on type relations and string matching. [1] uses crowdsourcing to detect quality issues. [26] uses external datasources and statistical data to detect problems in the knowledge base and fix them. Our work differs from all of these works in that we use the past edits as ground truth, and so are able to easily generalize over a large set of constraint types, instead of focusing on specific cases like *sameAs* or a wrong subject or object. Some works in completely different domains also use

the idea of learning repairs from past corrections. For example, [5] learns how to fix errors in source code based on previous error corrections.

**Graph neural networks.** Neural networks are often applied to knowledge base related tasks [30]. However, there does not seem to be any work in the literature that uses neural networks to predict repairs in knowledge bases. Some works use neural networks to reason on top of knowledge bases [15]. Several other works tackle the knowledge graph completion task by mining rules [13] or by link prediction. Our approach takes inspiration from these methods, but ultimately tackles a different problem: We do not want to predict links, but the correction of the violation of a constraint. We refer to [30] for a detailed survey about graph neural networks.

## 3 Preliminaries

Let us call $\mathcal{T}$ the set of all RDF terms (IRIs, blank nodes and literals). In our work, we see a *knowledge base* (KB) $\mathcal{K}$ as a set of triples of elements of $\mathcal{T}$. These triples are written $\langle s, p, o \rangle$, where $s$ is the subject of the triple (which cannot be a literal), $p$ the predicate (which has to be a IRI), and $o$ the object. We make the unique name assumption in what follows. If this assumption does not hold, and if the KB allows OWL entailment, then $a = b$ can be replaced by $\langle a, \texttt{owl:sameAs}, b \rangle$ in what follows without affecting the validity of our approach.

We define a *conjunctive query* (CQ) as a query of the form $\mathbf{C}(\boldsymbol{x}) = \langle s_1, p_1, o_1 \rangle \wedge \cdots \wedge \langle s_n, p_n, o_n \rangle$, where $\boldsymbol{x}$ is a sequence of variables, the $p_i$ are constants, and the $s_i$ and $o_i$ are variables over $\boldsymbol{x}$ or constants. We write $\mathcal{K} \models \mathbf{C}(\boldsymbol{x})$ iff there exists a binding for $\boldsymbol{x}$ such that $\langle s_1, p_1, o_1 \rangle \in \mathcal{K}$, ..., $\langle s_n, p_n, o_n \rangle \in \mathcal{K}$. We also define similarly a *disjunctive query* (DQ) as a query of the form $\mathbf{D}(\boldsymbol{x}) = \langle s_1, p_1, o_1 \rangle \vee \cdots \vee \langle s_n, p_n, o_n \rangle$ and we write $\mathcal{K} \models \mathbf{D}(\boldsymbol{x})$ iff there exists a binding for $\boldsymbol{x}$ and an $i$ such that $\langle s_i, p_i, o_i \rangle \in \mathcal{K}$. We also allow term equalities like $x = y$ and we write $\top$ (true) for the empty CQ and $\bot$ (false) for the empty DQ.

A KB can define *constraints*. Such constraints can enforce, e.g., that some information must be present (e.g. every person should have a birth place), or that some triple combinations may not occur (e.g. a person can have at most one birth place). Following [23], we express these constraints as rules. Such a rule takes the form $\varGamma(\boldsymbol{x}) : \mathbf{B}(\boldsymbol{x}) \to \exists \boldsymbol{y} \, \mathbf{H}(\boldsymbol{x}, \boldsymbol{y})$ where $\boldsymbol{x}$ and $\boldsymbol{y}$ are sequences of variables, $\mathbf{B}$ is a CQ and $\mathbf{H}$ a DQ. For example, the rule $\varGamma_1(x) : \langle x, \texttt{rdf:type}, \texttt{schema:Person} \rangle \to \exists y \, \langle x, \texttt{schema:birthPlace}, y \rangle$ says that all people must have a birth place. The rule $\varGamma_2(x, y_1, y_2) = \langle x, \texttt{schema:birthPlace}, y_1 \rangle \wedge \langle x, \texttt{schema:birthPlace}, y_2 \rangle \to y_1 = y_2$ says that entities can have at most one birth place. It is clear that any constraint with a combination of $\wedge$, $\vee$ and $\neg$ in the body can be translated into an equivalent conjunction of constraints in this formalism. In particular, the formalism can express constraints such as the following:

- Domain constraints. The rule to enforce that the domain of a property $p$ is $d$ is: $\varGamma(s, o) = \langle s, p, o \rangle \to \langle s, \texttt{rdf:type}, d \rangle$.

- Range constraints. The rule to enforce that the range of a property $p$ is $d$ is: $\Gamma(s, o) = \langle s, p, o \rangle \rightarrow \langle o, \texttt{rdf:type}, d \rangle$.
- Functional relations. The rule to enforce the functionality of a property $p$ is: $\Gamma(s, o_1, o_2) = \langle s, p, o_1 \rangle \wedge \langle s, p, o_2 \rangle \rightarrow o_1 = o_2$.
- Inverse functional relations. The rule to enforce the inverse functionality of a property $p$ is: $\Gamma(s_1, s_2, o) = \langle s_1, p, o \rangle \wedge \langle s_2, p, o \rangle \rightarrow s_1 = s_2$.
- Symmetric relations. The rule to enforce that $p$ is symmetric is: $\Gamma(s, o) = \langle s, p, o \rangle \rightarrow \langle o, p, s \rangle$.
- Conflicts between properties. The rule to enforce that $p_1$ and $p_2$ have disjoint domains is: $\Gamma(s, o_1, o_2) = \langle s, p_1, o_1 \rangle \wedge \langle s, p_2, o_2 \rangle \rightarrow \bot$.

Given a constraint $\Gamma(\boldsymbol{x}) : \mathbf{B}(\boldsymbol{x}) \rightarrow \exists \boldsymbol{y} \mathbf{H}(\boldsymbol{x}, \boldsymbol{y})$, a *violation* in a KB $\mathcal{K}$ is a binding for $\boldsymbol{x}$ such that (1) $\mathcal{K} \models \mathbf{B}(\boldsymbol{x})$ and (2) there is no $\boldsymbol{y}$ such that $\mathcal{K} \models \mathbf{H}(\boldsymbol{x}, \boldsymbol{y})$. For example, if the knowledge base is $\mathcal{K} = \{\langle \texttt{JohnDoe}, \texttt{rdf:type}, \texttt{schema:Person} \rangle\}$ then $\boldsymbol{x} = \texttt{JohnDoe}$ is a violation of $\Gamma_1$ in $\mathcal{K}$ because $\mathcal{K} \models \langle \texttt{JohnDoe}, \texttt{rdf:type}, \texttt{schema:Person} \rangle$ and $\nexists y\ \mathcal{K} \models \langle x, \texttt{schema:birthPlace}, y \rangle$.

The *violation triples* of a violation are the instantiated triples of the body of the constraint. In our previous example, the violation triples would be the single triple $\langle \texttt{JohnDoe}, \texttt{rdf:type}, \texttt{schema:Person} \rangle$.

To repair a violation, we need an *edit action*. Following [23], we define an edit action as a pair $(\mathcal{M}^+, \mathcal{M}^-)$ of one triple to add and one triple to remove. At most one of these triples can be absent, and we write $\emptyset$ for such a triple. The rationale for this representation is that, due to the disjunctive nature of the head of the constraint, the addition of a single triple can remove the violation, and due to the conjunctive nature of the body of the constraint, the removal of a single triple can also fix the constraint. Furthermore, a removal combined with an addition corresponds to a replacement, which is a frequent edit action in KBs. The *result* of an edit action $(\mathcal{M}^+, \mathcal{M}^-)$ on a KB $\mathcal{K}$ is the KB $\mathcal{K}' = (\mathcal{K} \cup \{\mathcal{M}^+\}) \setminus \{\mathcal{M}^-\}$ (omitting either set if the triple is absent in the edit action). An edit action is a *repair* of a violation of a constraint, if the violation no longer exists in the result of the edit action.

For example, the two possible edits to repair a violation of $\Gamma_1$ are to remove the triple $\langle x, \texttt{rdf:type}, \texttt{schema:Person} \rangle$ or to add a triple $\langle x, \texttt{schema:birthPlace}, y \rangle$ with a correct instantiation for $y$.

We aim at learning "good" repair edits that make the KB closer to the real world by adding "true" facts and removing "false" facts. Formally, a "good" edit is an edit $(\mathcal{M}^+, \mathcal{M}^-)$ such that $\mathcal{M}^+$ is absent or in $\mathcal{K}^i$, and $\mathcal{M}^-$ is absent or not in $\mathcal{K}^i$, where $\mathcal{K}^i$ is the "ideal" knowledge base representing the real world [25]. The *edit history* of a KB $\mathcal{K}$ is the sequence of edit actions that have been applied, starting from the empty KB, to yield $\mathcal{K}$. In this paper, we aim at building a predictor that takes as input the edit history of a KB and a set of constraints, and that predicts good repair edits in the sense described above.

## 4    Baseline Approach

[23] presents a rule learning approach for our problem, called *CorHist*. We introduce this system here briefly, before presenting our new system in the next section. CorHist takes as input the edit history of a KB and constraints. It mines correction rules of the form $\Gamma(\boldsymbol{x}) \wedge \langle s_m, p_m, o_m \rangle \wedge \langle s_c, p_c, o_c \rangle \rightarrow (\mathcal{M}^+(\boldsymbol{x}), \mathcal{M}^-(\boldsymbol{x}))$, where $\Gamma(\boldsymbol{x})$ is a constraint, $s_m$ and $s_c$ are variables in $\boldsymbol{x}$, $p_m$ and $p_c$ are constants, $o_m$ is a constant or a variable in $\boldsymbol{x}$, $(\mathcal{M}^+(\boldsymbol{x}), \mathcal{M}^-(\boldsymbol{x}))$ is the predicted edit action, and $o_c$ is a constant, a variable in $\boldsymbol{x}$ or a new free variable. $\langle s_c, p_c, o_c \rangle$ is optional. This correction rule means that if there exists a violation $\boldsymbol{x}$ of $\Gamma$ in $\mathcal{K}$, and if $\langle s_m, p_m, o_m \rangle(\boldsymbol{x}) \in \mathcal{K}$ and $\langle s_c, p_c, o_c \rangle(\boldsymbol{x}) \in \mathcal{K}$, then $(\mathcal{M}^+(\boldsymbol{x}), \mathcal{M}^-(\boldsymbol{x}))$ is a good repair. For example, CorHist is able to mine correction rules such as $\Gamma_1(x) \wedge \langle x, \mathtt{rdf:type}, \mathtt{schema:Person} \rangle \wedge \langle x, \mathtt{rdf:type}, \mathtt{schema:Place} \rangle \rightarrow (\emptyset, \langle x, \mathtt{rdf:type}, \mathtt{schema:Person} \rangle)$. This rule repairs violations of the constraint $\Gamma_1$ presented in Section 3 by stating that "if $x$ violates the constraint $\Gamma_1$ and is a $\mathtt{schema:Place}$, then the $\mathtt{schema:Person}$ type should be removed".

For this purpose, CorHist uses an adaptation of the AMIE algorithm [12]. It takes as input the past corrections dataset and the facts about the entities mentioned in the violation triples. It assumes that there is only one violation triple. It first generates simple rules by taking each past correction from the dataset, and by replacing constants by variables in both the violation triple and the associated correction. This leads to correction rules without context, like $\Gamma_1(x) \rightarrow (\emptyset, \langle x, \mathtt{rdf:type}, \mathtt{schema:Person} \rangle)$. Then, CorHist refines these correction rules by adding an extra triple to the rule body. For this, it looks for an extra possible pattern in the facts about the entities that are already used in the rule body, and adds this pattern to the rule body.

CorHist ranks the correction rules according to their confidence on the past corrections, i.e. the number of times the rule predicts the correct edit divided by how many times the rule predicts an edit. It also prunes the rules that have too low a confidence, predict less than 10 edits on the training dataset, or have extra triples in the body that do increase the confidence by at least 5%. When CorHist has to predict an edit, it uses the matching correction rule with the highest confidence. The minimal confidence threshold used for pruning is chosen so as to optimize the F1-score on a cross-validation dataset.

CorHist can be improved easily as follows:

- We allow multiple violation triples in the rule by applying exactly the same initialization step, but with multiple triples.
- We allow up to three additional triple patterns $\langle s_c, p_c, o_c \rangle$. These patterns must have a constant predicate and their subject must be a variable in $\boldsymbol{x}$. This allows more specialized rules than CorHist, which supports only one additional triple pattern.
- We allow the system to take into account information from external Web pages about the entities. This information can help the system choose, e.g., between two possible birth places, simply by checking if one of them appears on the Web page.

This last point works as follows: For every URL $s$ that appears in the violation triples, we add two new facts: one is $\langle s, \texttt{hist:pageStatusCode}, \texttt{XXXX}\rangle$, where XXXX is the HTTP response code of that URL. The other is $\langle s, \texttt{hist:pageContainsLabel}, o\rangle$, where $o$ is the object of a triple connecting $o$ to $s$, and one of $o$'s labels (found with the `rdfs:label` relation) appears in the HTML page of $s$. For example, if we consider the triple $\langle \texttt{Douglas\_Adams}, \texttt{schema:sameAs}, \texttt{<http://viaf.org/113230702>}\rangle$ from Wikidata, we fetch the external URL from `viaf.org`, and we add the triples $\langle \texttt{<http://viaf.org/113230702>}, \texttt{hist:pageStatusCode}, 200\rangle$ and $\langle \texttt{<http://viaf.org/113230702>}, \texttt{hist:pageContainsLabel}, \texttt{Douglas\_Adams}\rangle$, because the label `"Douglas Adams"@en` of the entity `Douglas_Adams` appears on the page.

   We call this improved system *CorHist+*. While we will see in our experiments that CorHist+ improves over CorHist, both systems still suffer from a systematic weakness: They can take into account only triples that explicitly appear in a correction rule – and not shallow signals from the state of the knowledge base. We will now see how to remedy this.

## 5   Approach

To overcome the limitation of the rule mining approach explored by CorHist, we design a new neural network architecture to implement a correction predictor. Our predictor takes as input the violated constraint, the violation triples, and the facts about the entities that are mentioned in the violation triples. It predicts as output a triple to add and/or a triple to delete. At training time, the input and the outputs come from the edit history of the KB. At prediction time, the input comes from the current state of the KB. In the following, we first present a "conversion" of CorHist+ to a neural network called *Bass-RL*, before adding new components to improve its performance, leading to our final *Bass* architecture[1].

### 5.1   Bass-RL

The goal of Bass-RL is to build a neural network that mirrors exactly the functioning of CorHist+. Figure 1 shows the basic architecture of our network. It is composed of 3 input components, which all feed into the "edit prediction" component. The dimensions of the internal layers are parametrized by a constant $d$, which is a hyperparameter of the network. The inputs are:

- **The constraint.** We give to each constraint a unique integer id and then we use a trained embedding matrix to create a $d$-dimensional vector from the one-hot encoding of the constraint id. This vector is then fed into the edit predictor. This vector gives to the network the information which type of constraint we aim to fix.

---

[1] The "Bass" name derives from "CorHist", because a bass is a singer with a deep voice, and "corhist" is an old English word for "singer".

– **The violation triples.** For each of the $k$ violation triples, we encode the predicate and the object by help of the "term embedding" component. These encodings are concatenated and fed into the edit prediction component. The subject of the violation triple is not used, because it cannot be a constant.
– **The facts about the entities** that appear in the violation triples. We encode only the predicates and the objects, because the subject is already known to the network. For example, if we consider the violation triple ⟨JohnDoe, schema:birthPlace, Paris⟩, the two entities are JohnDoe and Paris, and we embed their predicates and objects – i.e., {(rdf:type, schema:Person), (schema:gender, schema:Male), . . .} for JohnDoe and {(rdf:type, schema:Place), (schema:country, France), . . .} for Paris. There are $2k$ mentioned entities, 2 for each of the $k$ violation triples. Each fact is embedded using the "entity fact embedding" component. Then the output vectors are concatenated and fed into the edit prediction component.

Let us now describe each component in detail.

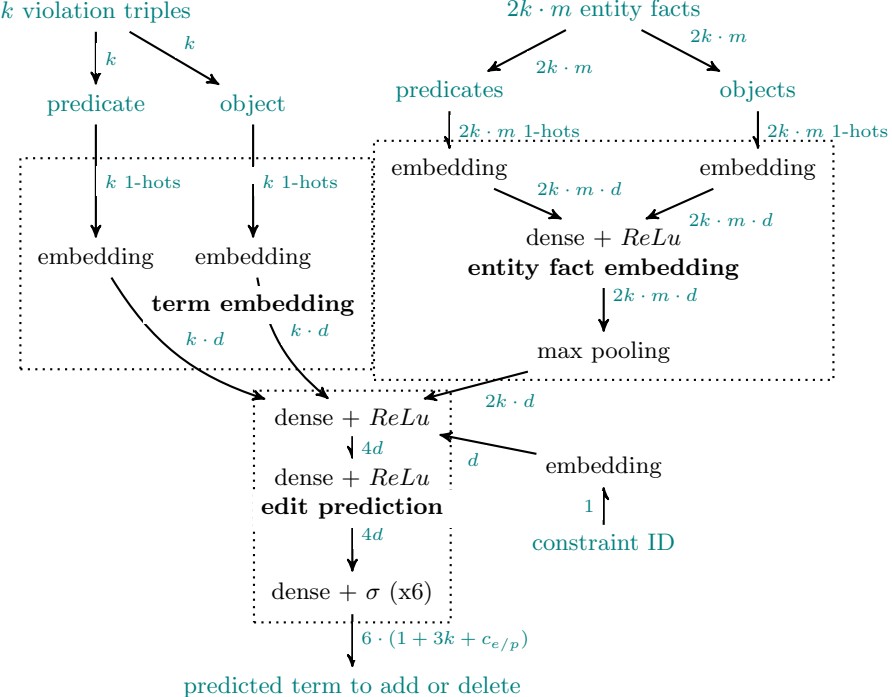

**Fig. 1.** Network architecture of Bass-RL. $d$ is the vector dimension hyperparameter, $k$ is the max number of violation triples, and $m$ is the max number of facts per entity.

**Term embedding.** We embed RDF terms as follows: predicates are one-hot encoded and then embedded into a space of dimension $d$ using an embedding

matrix $\mathcal{P}$. We embed similarly the objects using an matrix $\mathcal{E}$. These matrices $\mathcal{E}$ and $\mathcal{P}$ are shared by all object and predicate embedding operations. They are trained at the same time as the neural network. This embedding allows the edit predictor to act on specific predicates and entities in the violation triples.

**Entity fact embedding.** Similarly to CorHist, we give the neural network the facts about the $2k$ entities involved in the constraint violation. A classical approach for this purpose would be to use entity embeddings, i.e., to embed the entity itself. However, entity embeddings have two drawbacks: First, they require expensive pre-training. Second, and more crucially, they do not work with new entities. Therefore, we embed not the entity, but the facts that the entity is involved in. While the objects of these facts will still be encoded using learned embeddings (and thus cannot be entities that are unknown at training time), the subjects can be entities that have never been seen at training time. This allows the network to check constraints on newly added entities.

We encode the (predicate, object) facts of each entity mentioned in the violation triples as follows: we embed the predicates by reusing the same predicate embedding matrix $\mathcal{P}$ described previously. We embed similarly the objects by reusing the same entity embedding matrix $\mathcal{E}$. Then we combine the predicate and the object using a dense layer with a rectified linear unit (ReLU) non linearity[2]. Then we merge the obtained embeddings for each (predicate, object) using a max pooling layer to get a single vector for the entity. If the entity is a URL to an external website, we add the URL triples that we have introduced for CorHist+ (Section 4) to the set of (predicate, object) pairs.

**Edit prediction.** The edit prediction component takes as input the previous components, i.e. the constraint id embedding, the embeddings of the $k$ violation triple predicates and objects, and the embedding of the facts about the $2k$ entities mentioned in the violation triples. This data is then fed into a multi-layer perceptron. We use two hidden layers of dimensions $4d$ with the ReLU non-linearity.

Our network outputs two triples – one to add and one to delete. Each triple is given by its three components (subject, predicate, and object). Hence, our network has $2 \times 3$ output components. Each output component could of course just be the one-hot encoding of a predicate or entity. However, then the network would have to learn each instantiation of a constraint individually (as in "If the subject of the violation triple is Elvis, then the subject of the addition triple should be Elvis", "If the subject is Madonna, then..."). Therefore, we allow the network to output a code, as in "The subject of the addition triple is the object of the first constraint violation triple". To permit nevertheless the output of constants as well, we combine both approaches. With this, each of the 6 outputs ($2 \times 3$ triple components) works in the same way, classifying the output term into one of the following options:

---

[2] This function is defined as $ReLU(x) = max(0, x)$

– 1 class to state that the output is not existent (the triple should not be returned), or unknown. We call it the class 0.
– $3k$ classes to state that the output term is the same as the subject/predicate or object of one of the $k$ violation triples. For example, the class 1 corresponds to the subject of the first violation triple, the class 2 to its predicate, the class 3 to its object, the class 4 to the subject of the second violation triple etc.
– $c_p$ or $c_e$ classes to state that the output is one of the $c_p$ predicates (for the predicate to add or delete outputs) or one of the $c_e$ entities (for the other outputs) from a the list of predicates/entities found at least $t$ times in the expected outputs from the training data (where $t$ is a hyperparameter).

This leads to $1 + 3k + c_e$ possible classes (and so, output neurons) for each of the four subject/object outputs and $1 + 3k + c_p$ for the two predicate outputs. Each of these outputs are implemented like regular classifiers using a dense layer with a softmax non-linearity[3]. The outputs returns a $1 + 3k + c_{e/p}$ dimensional vector. The $i$th vector output is the predicted probability of $i$th class.

To retrieve the final edit, we consider the output for both the "delete" triple and the "add" triple. If the three outputs for the subject, predicate and object of the triple give known values (a known entity or predicate or a known violation triple term given in the input) we build the triple to add or to delete from these outputs. If the three outputs give the "not defined" class, we return no triple. This means that the edit does not contain a triple to add, or a triple to delete, respectively. If there is only one or two components returning "not defined", we assume that the network has not been able to predict a correction.

## 5.2   Bass

We now present our improved repair predictor, *Bass*. Figure 2 shows the architecture of our network. We designed two improvements over Bass-RL:

**RDF literal embedding.** We want our predictor to be able to act on RDF literals. For example, if every book can have at most one ISBN number, and if a given book has two ISBNs, `2-7654-1005-4` and `abc`, it is easy to decide that `abc` should be abandoned, if we allow the network to access the literals.

For this purpose, we first remove the datatype IRI. This leads to strings like `"Elvis Presley"@en` for a language tagged string or `"42"` for an integer. Then, we tokenize the string with the BERT tokenizer [10] and apply an embedding matrix on each token. We then apply a max pooling on the embedding sequence to get another $d$-dimensional vector. We use the BERT tokenizer to provide a better support for out-of-vocabulary words. It also allows to better handle complex values like numbers and dates that a regular tokenizer based on a whitespace and/or other characters split. It also allows keeping more significant elements than a simple char based encoding. This input is not used when the objects are IRIs or blank nodes.

---

[3] $\sigma(x_i) = \frac{e^{x_i}}{\sum_{j=1}^{D} e^{x_j}}$ for all $i \in 1, \dots, D$ if there are $D$ possible classes.

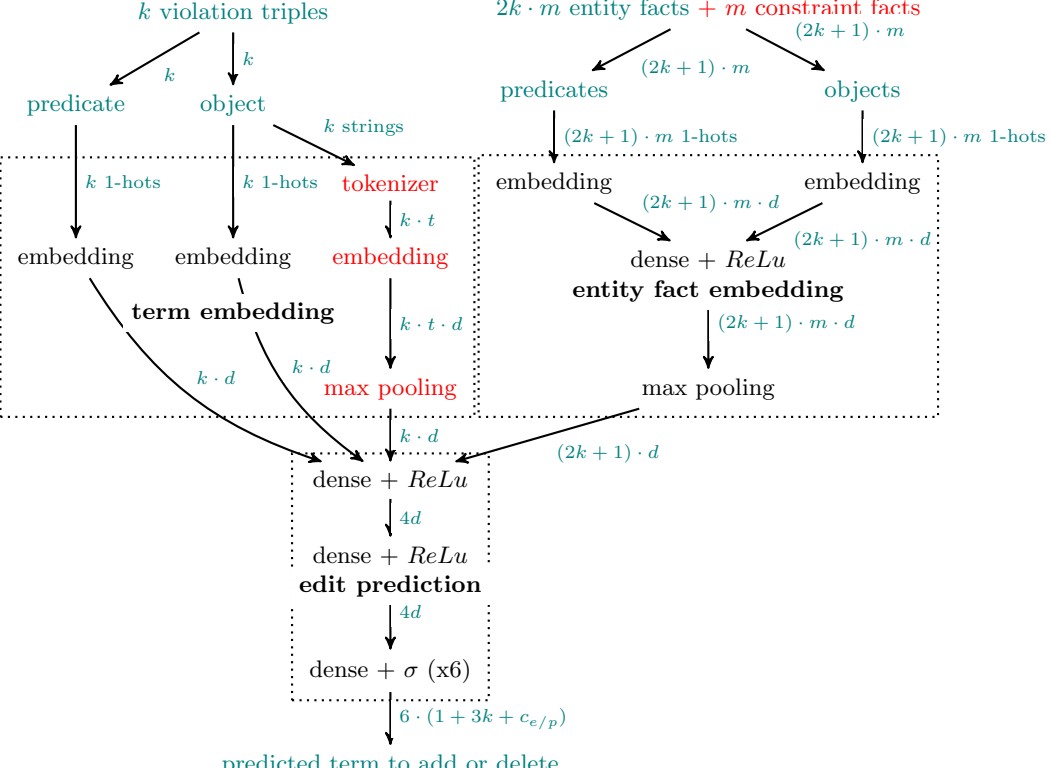

**Fig. 2.** Bass network architecture when there is just one violation triple. $d$ is the vector dimension hyperparameter, $k$ the max number of violation triples, $m$ the max number of facts per entity, and $t$ the max number of string tokens. Additions compared to Bass-RL are in red.

**Constraint embedding.** Bass-RL, like CorHist, encoded a constraint by an ID. This does not allow the network to generalize over the constraints, to treat similar constraints similarly, or to learn new constraints. To remedy this shortcoming, we encode the constraint by a set of (predicate, object) pairs and we encode this set using the entity fact embedding component introduced earlier.

To encode the constraint by a set of (predicate, object) pairs, we rely on *constraint shapes*. The constraint shape of a constraint $\Gamma$ is a constraint $\Gamma^s$ where all constants in $\Gamma$ have been replaced by fresh variables. For example, the shape of the constraint $\Gamma_1(x) : \langle x, \mathtt{rdf{:}type}, \mathtt{schema{:}Person} \rangle \rightarrow \exists y\ \langle x, \mathtt{schema{:}birthPlace}, y \rangle$ is $\Gamma_1^s(x) : \langle x, p_1, o_1 \rangle \rightarrow \exists y\ \langle x, p_2, y \rangle$. Two constraint shapes are equivalent if they have the same components up to a renaming of variables. With this definition, we encode a constraint $\Gamma(\boldsymbol{x}) : \varphi(\boldsymbol{x}) \rightarrow \exists \boldsymbol{y}\ \varphi_1(\boldsymbol{x}, \boldsymbol{y}) \vee \cdots \vee \varphi_n(\boldsymbol{x}, \boldsymbol{y})$ by the following set of property value pairs:

- $(\mathtt{bass{:}constraintShape}, i)$ where $i$ is an identifier assigned to the equivalence class of the constraint shape $\Gamma^s$.

- $(p_i, o_i)$ for each $\langle s_i, p_i, o_i \rangle$ in $\varphi$ and $\varphi_i$ where $o_i$ is a constant.
- $(\hat{p}_i, s_i)$ for each $\langle s_i, p_i, o_i \rangle$ in $\varphi$ and $\varphi_i$ where $s_i$ is a constant and $\hat{p}_i$ is the inverse property of $p_i$.[4]

These components give us our new Bass network.

## 6  Experiments

### 6.1  Dataset

We evaluated our algorithm on Wikidata [28]. We considered the same 10 kinds of constraints as in [23]. The dataset provided by [23] contains, for each past correction, the violated constraint and a single violation triple. Thus, running CorHist required access to the Wikidata edit history, so as to extract the other violation triple for "conflict with", "single", and "distinct" constraint types and to retrieve the facts about the entities mentioned in the violation triple. To simplify the re-use of our dataset, we added these items to the data, so that CorHist and Bass can now be trained without having to access the edit history. We also included the content of the external Web pages in the dataset, so that CorHist+ and Bass (and future approaches) do not need to download the pages again.

Different from [23], we limited the number of past corrections to 200k per constraint type (except for "one of" constraints, which have only 23k extracted past corrections in the full dataset). This limitation is made to facilitate the use of the dataset (the previous one had a size of 36GB), and to allow fetching all mentioned Web pages in a week. We split the dataset into 80% training set, 10% cross-validation set, and a 10% test set. This improved dataset is publicly available on FigShare[5].

### 6.2  Systems

**Implementation.** We implemented Bass with the Keras API of Tensorflow 2. For the BERT tokenization, we use the HuggingFace "torkenizers" library [29]. Our implementation is publicly available on GitHub[6]. With $\mathcal{E}$ and $\mathcal{P}$, we embed only the entities and predicates with at least 100 occurrences. We do the same for the output by setting $t = 100$. We choose to set all the embedding sizes to $d = 128$. We use the constraint type identifier ("single value", "value type"...) to identify the constraint shape, and we use the Wikidata statement that encodes the constraint to generate the (predicate, object) triples that describe the constraint.

---

[4] Using a new IRI if there is no inverse property of $p_i$ already in the KB.

[5] https://doi.org/10.6084/m9.figshare.13338743

[6] https://github.com/Tpt/bass-materials

**Table 1.** Evaluation of the correction rules mined by Bass, CorHist [23] and CorHist+ and comparison with the baselines. Best F scores in bold.

| Constraint type | | Micro average | | | Macro average | | |
|---|---|---|---|---|---|---|---|
| | | Prec. | Rec. | F | Prec. | Rec. | F |
| Type | add | 0.53 | 0.17 | 0.26 | 0.28 | 0.10 | 0.14 |
| | delete | 0.04 | 0.04 | 0.04 | 0.08 | 0.08 | 0.08 |
| | CorHist | 0.88 | 0.62 | 0.73 | 0.96 | 0.28 | 0.43 |
| | CorHist+ | 0.86 | 0.75 | 0.80 | 0.89 | 0.34 | 0.49 |
| | Bass | 0.92 | 0.79 | **0.85** | 0.83 | 0.36 | **0.50** |
| Value type | add | 0.20 | 0.07 | 0.10 | 0.35 | 0.11 | 0.16 |
| | delete | 0.01 | 0.01 | 0.01 | 0.04 | 0.04 | 0.04 |
| | CorHist | 0.70 | 0.62 | 0.66 | 0.86 | 0.35 | 0.50 |
| | CorHist+ | 0.70 | 0.63 | 0.66 | 0.81 | 0.43 | **0.56** |
| | Bass | 0.78 | 0.69 | **0.73** | 0.70 | 0.28 | 0.40 |
| One-of | delete | 0.27 | 0.27 | 0.27 | 0.43 | 0.43 | 0.43 |
| | CorHist | 0.84 | 0.72 | **0.78** | 0.84 | 0.34 | **0.48** |
| | CorHist+ | 0.84 | 0.72 | **0.78** | 0.84 | 0.34 | **0.48** |
| | Bass | 0.86 | 0.71 | **0.78** | 0.77 | 0.26 | 0.39 |
| Item requires statement | add | 0.99 | 0.11 | 0.20 | 0.92 | 0.13 | 0.22 |
| | delete | 0.02 | 0.02 | 0.02 | 0.07 | 0.07 | 0.07 |
| | CorHist | 0.94 | 0.30 | 0.46 | 0.98 | 0.17 | 0.29 |
| | CorHist+ | 0.85 | 0.36 | **0.51** | 0.94 | 0.19 | **0.32** |
| | Bass | 0.89 | 0.35 | 0.50 | 0.76 | 0.17 | 0.28 |
| Value requires statement | add | nan | 0 | nan | nan | 0 | nan |
| | delete | 0.02 | 0.02 | 0.02 | 0.09 | 0.09 | 0.09 |
| | CorHist | 0.96 | 0.64 | 0.77 | 0.95 | 0.33 | 0.49 |
| | CorHist+ | 0.90 | 0.69 | 0.78 | 0.89 | 0.39 | **0.55** |
| | Bass | 0.98 | 0.75 | **0.85** | 0.72 | 0.32 | 0.44 |
| Conflict with | delete | 0.39 | 0.39 | 0.39 | 0.44 | 0.44 | 0.44 |
| | CorHist | 0.93 | 0.47 | 0.63 | 0.91 | 0.36 | 0.51 |
| | CorHist+ | 0.87 | 0.84 | 0.86 | 0.83 | 0.46 | 0.59 |
| | Bass | 0.91 | 0.86 | **0.88** | 0.77 | 0.71 | **0.74** |
| Inverse + Symmetric | add | 0.91 | 0.91 | 0.91 | 0.82 | 0.82 | 0.82 |
| | delete | 0.07 | 0.07 | 0.07 | 0.11 | 0.11 | 0.11 |
| | CorHist | 0.95 | 0.91 | 0.93 | 0.91 | 0.72 | 0.80 |
| | CorHist+ | 0.94 | 0.92 | 0.93 | 0.90 | 0.73 | **0.81** |
| | Bass | 0.97 | 0.94 | **0.95** | 0.87 | 0.58 | 0.69 |
| Single value | delete | 0.45 | 0.45 | 0.45 | 0.42 | 0.42 | 0.42 |
| | CorHist | 0.85 | 0.26 | 0.39 | 0.90 | 0.10 | 0.18 |
| | CorHist+ | 0.55 | 0.50 | 0.53 | 0.74 | 0.23 | 0.36 |
| | Bass | 0.74 | 0.64 | **0.69** | 0.60 | 0.52 | **0.56** |
| Distinct values | delete | 0.55 | 0.55 | 0.55 | 0.45 | 0.45 | 0.45 |
| | CorHist | 0.61 | 0.53 | 0.56 | 0.90 | 0.16 | 0.27 |
| | CorHist+ | 0.58 | 0.57 | **0.57** | 0.80 | 0.26 | 0.39 |
| | Bass | 0.59 | 0.56 | **0.57** | 0.48 | 0.43 | **0.46** |
| Total | add | 0.46 | 0.14 | 0.22 | 0.33 | 0.11 | 0.16 |
| | delete | 0.24 | 0.24 | 0.24 | 0.22 | 0.22 | 0.22 |
| | CorHist | 0.83 | 0.53 | 0.65 | 0.94 | 0.23 | 0.37 |
| | CorHist+ | 0.75 | 0.64 | 0.69 | 0.85 | 0.30 | 0.44 |
| | Bass-RL | 0.77 | 0.65 | 0.70 | 0.64 | 0.34 | 0.44 |
| | Bass | 0.80 | 0.68 | **0.73** | 0.69 | 0.39 | **0.49** |

**Training.** We trained Bass on the training set for all constraint types at the same time, using the sum of categorical cross-entropy loss[7] for the 6 classification outputs and the Adam [16] optimizer. To ease the training we used the validation set to keep the best epoch according to the loss against the cross-validation dataset. We trained the model for 6 epochs on the full training dataset with a mini-batch size of 256, the best model being found after the 3rd epoch. After loading the dataset into memory, training took 18min using a laptop with an Nvidia Quadro P3000 mobile GPU, an Intel Core i7-7700HQ CPU, and 32GB of RAM. The rather large batch size was chosen so as to improve the training stability. We experimented with larger batch sizes, but these do not improve the performances.

**Baselines.** We compare our approach against *CorHist* and *CorHist+* described in Section 4, and the two baselines from [23]. For CorHist+, we use a minimal support of 10 and a minimal confidence between 0.1 and 1.

The two baselines *delete* and *add* are basic ones without any learning: *Delete* uses the fact that all Wikidata constraint violations have a "focus" on a single existing triple whose removal would remove the violation. Our baseline just removes this triple. *Add* tries to add a new triple to solve the constraint violation. For "inverse" and "symmetric" constraints, this baseline adds the missing reverse triple and performs very well. For "item requires statement", "value requires statement", "Type" and "Value type", it adds a possibly missing triple only if it is possible to know the expected value(s) from the constraint rule by picking one of the expected values randomly.

### 6.3   Results

Table 1 compares the performances of *Bass*, *CorHist+*, *CorHist* and the two baselines using our dataset test set. To counter the non-deterministic nature of gradient descent based training, we ran Bass training multiple times. Its performances were stable enough to not change the performance ranking. The other approaches, CorHist(+) and the baselines, are deterministic.

As shown in the evaluations, Bass significantly outperforms CorHist+, which itself significantly improves over CorHist. Indeed, we have conducted a Wilson score test at confidence 95%, and confirm that the confidence interval is of size 0.2% for all approaches – an order of magnitude lower than the score gaps between the different methods. The strongest improvements are seen for the constraint types "Conflict with" and "Single value", which both concern the removal of one value between two choices. CorHist+ significantly improves over CorHist (7% in macro average F-score) by allowing more complex rules, allowing more relevant decisions and integrating metadata from Web pages. The simple move from a rule learning algorithm (CorHist+) to a neural network (Bass-RL) provides a small performance improvement of 1% in micro average F-score. We

---

[7] The categorical crossentropy is defined as $CE(x) = \sum_{i=0}^{n} t_i log(p_i)$ where $t_i$ is the expected prediction for the class $i$, and $p_i$ is the predicted output.

believe that this improvement stems from the fact that, different from rules, neural networks are able to draw holistic conclusions from the context facts. The addition of textual data and of a structure representation of the constraint, which are hard to take into account with a rule mining approach, allows Bass to outperform Bass-RL by 5% in macro average F-score and by 3% in micro average, suggesting that textual data and a structured representation of the constraints help solving violations of under-represented constraints.

Bass takes around 18 minutes to train, whereas CorHist took 4 hours. Thus, the increase F1 provided by Bass actually comes with a decrease of the training time.

**Table 2.** Ablation study results.

| Approach | Micro average | | | Macro average | | |
|---|---|---|---|---|---|---|
| | Prec. | Rec. | F | Prec. | Rec. | F |
| Bass | 0.80 | 0.68 | **0.73** | 0.69 | 0.39 | **0.49** |
| Bass-RL | 0.77 | 0.65 | 0.70 | 0.64 | 0.34 | 0.44 |
| Bass minimal | 0.65 | 0.53 | 0.58 | 0.50 | 0.25 | 0.33 |
| Bass without object literals | 0.77 | 0.65 | 0.71 | 0.66 | 0.36 | 0.47 |
| Bass without entity facts | 0.70 | 0.59 | 0.64 | 0.54 | 0.31 | 0.39 |
| Bass without constraint | 0.77 | 0.65 | 0.71 | 0.60 | 0.35 | 0.44 |
| Bass with one hidden layer edit predictor | 0.79 | 0.67 | 0.72 | 0.69 | 0.38 | 0.49 |
| Bass with constraint ids | 0.80 | 0.68 | 0.73 | 0.66 | 0.37 | 0.47 |
| Bass with BiLSTM literals | 0.78 | 0.66 | 0.71 | 0.65 | 0.37 | 0.47 |
| Bass with entity facts attention | 0.76 | 0.65 | 0.70 | 0.63 | 0.36 | 0.46 |
| CorHist+ | 0.75 | 0.64 | 0.69 | 0.85 | 0.30 | 0.44 |
| CorHist | 0.83 | 0.53 | 0.65 | 0.94 | 0.23 | 0.37 |
| Deletion baseline | 0.24 | 0.24 | 0.24 | 0.22 | 0.22 | 0.22 |
| Addition baseline | 0.46 | 0.14 | 0.22 | 0.33 | 0.11 | 0.16 |

### 6.4   Ablation study

To understand the contribution of each component of our network, we remove the components one by one, and measure the performance. We also added Bass-RL and another variant, *Bass minimal*. This variant removes from Bass-RL the constraint embedding and the entity fact embedding. This leads to a network where the edit prediction component receives only the violation triple predicates and objects, embedded with $\mathcal{P}$ and $\mathcal{E}$.

Additionally to the ablation, we investigate some possible variants of our network:

- Replace the constraint description input by the Bass-RL constraint ID encoding. This gives the exact constraint to the model, without the possibility to generalize from the constraint description.
- Replace the max pooling layer in the literal embeddings by a bidirectional long short-term memory network (BiLSTM) [14].
- Replace the max pooling layer that aggregates the embeddings of the (predicate, object) tuples of the involved entities by an attention layer. We define

the attention following [19] as $\sigma(q \cdot V^\top) \cdot V$, where $\sigma$ is the softmax function. This attention layer uses for query $q$ the constraint embedding.

*Discussion.* Using the description of the constraint instead of the constraint ids does not change the micro average score, but increases the macro average scores. This means that the change helps for constraints with only a few past corrections, which have a weaker weight in the micro average. We thus observe a better generalization with respect to the constraints. Embedding the object literal values brings 2% of improvement on both F-scores, suggesting it brings some value on the 13% of violations where one of the objects is a literal that is not a date or a geographical coordinate. The addition of a bidirectional LSTM and attention would actually be detrimental to the performances of our network.

Overall, our experiments show that our architecture is well-designed, and that it is able to outperform the rule mining approach from [23] by a considerable margin.

## 7   Conclusion

In this paper, we have presented an approach to automatically repair constraint violations in Wikidata. Our experiments with various sets of constraint types show that our new approach provides significant improvements over the rule-based state of the art, by taking advantage of neural networks. This improvement stems from two factors: First, we find that the neural network can better take into account the shallow context of an entity. Second, the network can take advantage of other input, such as textual values, which is hard to exploit in traditional rule learning approaches.

For future work, we are considering to use the textual information contained in the knowledge base (labels, descriptions) and in the related Web pages. We also plan to update the user study done in [23] using Bass instead of CorHist. It might also be interesting to investigate how well CorHist and Bass perform when less training data is available, and to compare them with other static approaches for KB repairs.

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
