# OpenReview forum: "Neural Knowledge Base Repairs"
_eswc-conferences.org/ESWC/2021/Conference/Research_Track — ESWC 2021 Research_

### Official Review · AnonReviewer2 · 2021-01-11
**New versions of the edit history based Wikidata repair system CorHist**

**Rating:** 1
**Confidence:** 4
**Impact:** 4
**Design And Technical Quality:** 3

**Review:**

This paper presented three new versions towards their original system CorHist which repairs Wikidata constraint violations by utilising historical edit records. The first version is CorHist+ which takes the html page (i.e., URI in a triple) into consideration. The second version Bass-RL is a "conversion" of CorHist+ to a neural network which predicts the repair actions by the context including the term embedding and the entity fact embedding. The third version Bass augments Bass-RL by utilising more context including the literal and the constraint embedding.

Strong Points:
1) The topic is very meaning for improving the quality of Wikidata
2) The new versions improve the performance of the original version in [23] on the same task
3) The evaluation is extensive and data/codes are open

Weak Points:

The presentation can be further improved for higher readability. The descriptions to some technical details are ambiguous. It seems the authors (in some degree) assume the reader has some knowledge of the previous work [23]. More preliminaries, such as the background of Wikidata edit history and constraint shape (used in Section 5.2), would be helpful to understand the paper. In some degree I can understand this would exceed the 16-page size limit.

In the final paragraph, what does "a good edit is an edit $(\mathcal{M}^+, \mathcal{M}^+)$ such that $\mathcal{M}^+$ is absent or in $\mathcal{K}^i$, and $\mathcal{M}^-$ is absent or in $\mathcal{K}^i$" mean? In the definition of an edit, it uses $\mathcal{M}$ without any parameter, but in its usage, it directly uses $\mathcal{M}(x)$ with the input of $x$.

In introducing CorHist+, the insight of these operations for processing the URI and getting URI triple is not clear.

In the end of Page 7, in the statement "predicates are one-hot encoded and then embeded into a space of dimension of d using a trained embedding matrix $\mathcal{P}$", what does "trained embedding" mean? In my understanding, it is initially one-hot encoded and then trained during learning, instead of being pre-trained. Is it?

In Section 6.2, why only entities and predicates with >= 100 occurrences considered? Any reasons for this number?

The NN would predict a score to indicate whether an add/delete edit is needed or not. How the final decision is made? How the threshold is determined if it is based on a threshold? Otherwise Precision and Recall cannot measured. Is one threshold used in producing all the results in Table 1 and 2? These should be explained in the experiment setting.

Finally, any discussions of the proposed repair techniques to other knowledge bases beside Wikidata?


**Anonymity:**

Yes, I would like my review to remain anonymous.

**Reuse And Availability:**

4: High

**Subreviewer:**

I submitted this review.

---

> ### Author Rebuttal · Authors · 2021-01-28
>
> Thank you so much for your thoughtful review.
>
> > In the final paragraph, what does "a good edit $(M^+,M-)$ is an edit such that $M^+$ is absent or in $K^i$, and $M^-$ is absent or in $K^i_n$" mean? In the definition of an edit, it uses $M$ without any parameter, but in its usage, it directly uses $M(x)$ with the input $x$.
>
> This is because in the definition we are looking at an instantiated edit but CorHist rules build edits by extracting some variables from the rule bodies. Hence the $M(x)$.
>
> > In the end of Page 7, in the statement "predicates are one-hot encoded and then embeded into a space of dimension of d using a trained embedding matrix", what does "trained embedding" mean? In my understanding, it is initially one-hot encoded and then trained during learning, instead of being pre-trained. Is it?
>
> Yes, it is. We will remove the misleading "trained" word.
>
> > In Section 6.2, why only entities and predicates with >= 100 occurrences considered? Any reasons for this number?
>
> It is a threshold that we chose empirically to avoid a too large embedding matrix, which would make training the network more difficult.
>
> > The NN would predict a score to indicate whether an add/delete edit is needed or not. How the final decision is made? How the threshold is determined if it is based on a threshold? Otherwise Precision and Recall cannot measured. Is one threshold used in producing all the results in Table 1 and 2? These should be explained in the experiment setting.
>
> The way the neural network emits an edit is explained in detail on page 9: we consider the network not able to predict an edit if, for the add or the delete triple, one or two of the triple term classifier outputs "the triple should not exist" and the other classifiers outputs a value (i.e. they disagree).
>
> > Finally, any discussions of the proposed repair techniques to other knowledge bases beside Wikidata?
>
> If some constraint system and edit history is available then we believe CorHist and bass should be applicable. We shall add this thought as future work to our paper.
>
> Thank you again!

---

### Official Review · AnonReviewer1 · 2021-01-12
**Interesting work with some weak points that need to be addressed**

**Rating:** 1
**Confidence:** 4
**Impact:** 3
**Design And Technical Quality:** 4

**Review:**

This work continues a research line that focuses on fixing errors in RDF knowledge bases (called "repairs" by the authors, see minor comment below regarding this) using edit histories to learn how to do so. This work incorporates a machine learning component to the model, allowing for new cases to be handled and potentially better performance to be obtained.
The paper is generally well written, and addresses an interesting and relevant problem with potentially high impact. There are a few aspects that need to be addressed:

-- Motivation is rather lacking, since the authors do not discuss how the tool would best be used in practice. Under what conditions and in what manner would it be deployed in the real world?

-- In the introduction, the authors refer to [23] having a user study to evaluate agreement between humans and the suggestions made by the tool. Why was this not also included in this paper? It was not even mentioned as a necessary next step.

-- Analyzing Table 1 closely, one can see that the relationship between the precision and recall scores of the CorHist and the new approaches suggests that the latter are not "always better". In most cases, the typical tradeoff between the two comes up, which raises the question of what would happen if, for instance, the domain were of the kind in which having high precision is preferable to having overall better performance (i.e., focusing on F1 as the authors do). Some discussion regarding this is necessary, since otherwise the analysis is too one-dimensional. This comment is related to the first one, since motivation and limitations are often closely tied together.

Minor comments:

-- The title should be more specific, making it clear that the work focuses on RDF KBs and not more general ones.

-- Since the term "repair" is used in a rather large research area called consistent query answering born in the Databases community and later continued in the Ontology Languages area, it would be good for the authors to clarify that their approach is not directly related to these research lines.

-- Page 13:

"counter non-deterministic nature" --> "counter the non-deterministic nature"

"significantly outperform" --> "significantly outperforms"

-- Page 15: "with a various" --> "with various"

*** AFTER REBUTTAL:

I acknowledge having read the authors' response to my review.

**Anonymity:**

Yes, I would like my review to remain anonymous.

**Reuse And Availability:**

4: High

**Strong Points:**

Well written and practical results obtained on an interesting and relevant problem.

**Subreviewer:**

I submitted this review.

**Weak Points:**

Motivation and limitations, discussion of how results relate to previous work, and misleading title (see review).

---

> ### Author Rebuttal · Authors · 2021-01-28
>
>
> Thank you so much for your thoughtful review.
>
> This work is an improvement of the system presented in [23]. The work of [23] is already deployed in a live tool that suggests corrections to the Wikidata contributors. Our work is an improvement of [23], and it can replace the back-end of that tool transparently. we will note this in the paper.
>
> Indeed, a user study is the logical next step. We have not yet done it, because the user study of [23] validated the approach as such, and our current submission merely improves the precision and recall values. Nevertheless, we will mention this step as future work.
>
> About the precision/recall compromise, indeed, CorHist is often better at precision. However, the motivation of this work is the same as [23], i.e. suggesting corrections to apply. Since the receiver is thus the user rather than the knowledge base, we rather optimized for F1. The work "Steven C Funk and K Laurie Dickson. 2011. Multiple-choice and short-answer exam performance in a college classroom." cited by [23] suggests that a 40% precision rate is already enough to provide a useful suggestion tool -- a requirement that we beat by a large margin.
>
> We will also elaborate a bit more on the relation of our work to the classical repair methods as used in Description Logics and Databases.
>
> Thank you again!

---

### Official Review · AnonReviewer4 · 2021-01-13
**interesting neural based approached to learning to repair knowledge bases**

**Rating:** 2
**Confidence:** 4
**Impact:** 3
**Design And Technical Quality:** 4

**Review:**

This paper looks at model to repair knowledge bases by learning from edit distances. It does this by developing a new neural network architecture that mimics and then extends the authors existing work that used rule mining to tackle the same problem. The approach shows improved performance on knowledge base repair for wikidata. Importantly, the paper does a good job of building up the model showing an analogous neural architecture and then discussing extensions. There is an extensive ablation study and the authors do a good job of making an updated dataset and code available. Overall, I enjoyed reading the paper and I think the study itself is a solid contribution.

I did have a couple of concerns that could be addressed better.

## Baseline algorithms
First, the baseline comparisons are to the authors own work and some simple baselines. I think it would have been nice to have compared to a baseline that does not require the use of the edit history (e.g. Chen 2020) or at least make a stronger argument that the difference here is important, which leads me to the second area of concern.

## Datasets and around of edit history
A downside to the paper is that it only considers wikidata. It would have been nice to see this on other knowledge bases. Is it the case that other KBs don't have a similar edit history? At a minimum, it would have been nice to see an argument as to why only wikidata could be used. I could imagine that the edit history is easier to access but this would have been good to clairify.

Additionally, I would have liked to see a study on the impact of the amount of edit history needed. The paper said 200k edits were used what does performance look like as the number of edits decreases. This is important as it says something about the quantity of hand editing needed before the method can be applied.

Those were my major concerns. There are a couple of places where the paper could be improved in terms of explanation.

- On pg. 8. it says the output is a code that says "The subject of the additional triple is the object of the first constraint aviation triple" I really had a hard time trying to figure out the exact output of the neural network. Maybe an example of one vector of the output could be given

- In section 5. 1 you give an id to each constraint. Are the constraints just the ones enumerated in the preliminaries or where do they come from. This was somewhat unclear.

- How do you determine if an entity appears on the web page and how is this integrated into the neural network pipeline?

## Minor comment
- I would recommend not to use references in the abstract. Maybe just describe the method: "We build on an existing rule mining approach" or something like that.


__After Rebuttal__
Thanks for the clarifications. I think they'll make the paper better.



**Anonymity:**

Yes, I would like my review to remain anonymous.

**Reuse And Availability:**

5: Very High

**Strong Points:**

* State of the art performance for knowledge base repair
* Comprehensive ablation studies and multiple architectures
* New dataset
* Well written.

**Subreviewer:**

I submitted this review.

**Weak Points:**

* only one dataset (Wikidata)

---

> ### Author Rebuttal · Authors · 2021-01-29
>
> Thank you so much for your thoughtful review.
>
> We agree that it would be nice to compare our approach with other "static" approaches. It would also be interesting to investigate the impact of the amount of edit history. We shall investigate this in future work, and we will also note this as important future work in the paper.
>
> We will also improve the  wording of the neural network output description section, as you suggested. The neural network output is 6 classifier outputs, one per triple term (subject of the triple to add, predicate of the triple to add, ..., object of the triple to delete).
> Each of these classifiers is implemented by $o$ outputs, each representing a different class. Hence the neural network has $6o$ outputs in total. The class with the higher score is used. The possible classes are presented at the beginning of page 9. There is one class to state "this term is part of a triple that should not be in the output (in case the edit is just one addition or one removal)", $3k$ classes to pick a term in one of the $k$ violation triples in the output and $c_p$ or $c_e$ classes to choose a "well known" predicate or entity (to support e.g. "add the human type").
>
> What have been enumerated in the preliminaries are the constraint *types*. The constaints themselves come from Wikidata. Each constraint in Wikidata has a unique identifier (the identifier of the statement that encodes it).
>
> We determine if an entity appears on the web page by checking if the HTML of the page contains any of the entity labels (by string inclusion). We provide this information to the neural network by adding an extra triple to the web page "context facts". It is the exact same triples that are added for CorHist+ (cf. p. 6).
>
> Thank you again!

---

### Official Review · AnonReviewer3 · 2021-01-14
**A good paper with a useful novel approach and extensive experimental analysis**

**Rating:** 2
**Confidence:** 2
**Impact:** 3
**Design And Technical Quality:** 4

**Review:**

The authors are trying to improve the automated correction of constraint violations in knowledge graphs.  The approach considers far more features of the knowledge graph than just “the data captured by logical rules” including the edit history of rules that solved past violations.

The paper is well structured, there are two helpful figures in the “Approach” section describing the neural networks used to predict the repairs and two tables in the “Experiments” section comparing results, very comprehensively, of the authors own models with baselines and previously published models, as well as results assessing component importance in their own models via an ablation study.  The paper also includes links to download the dataset the authors created to train and test the models, as well as a github repository containing the python code for training the model.

The authors create three models, CorHist+, Bass-RL and Bass that incrementally improve on the previous model and have been created to improve on the current state of the art, CorHist.  The most interesting novel ideas are the use of feature embeddings to represent RDF terms, RDF literals and constraints via both ID and the “shape” of a constraint.  These embeddings are then passed into a neural network that outputs candidate constraint rules to add and/or delete.  Previously the CorHist rule mining approach only generated a single correction rule at a time, when the authors approach can identify pairs of add and delete correction rules.

The authors give a detailed description of the experiments used to validate the performance of the models and the impact of each of the features. They also briefly mention the direction of future work.

We acknowledge the rebuttal. It would be great if the authors could include the two answers in the next iteration of the paper.


**Anonymity:**

Yes, I would like my review to remain anonymous.

**Reuse And Availability:**

4: High

**Strong Points:**

- The approach uses “constraint shapes” to make the grouping of constraints more invariant to specific triple values/variables in the constraints.
- A detailed description of improvements to the previous training dataset, the implementation of the models and the specification of the training machine.
- The two figures that describe two of the neural networks implemented in conjunction with the text make it easy to understand how the hyperparameters impact the architecture of the network.
- The new dataset generated from this study reduces the previous need to access external web content to build the training set which will speed up research in this area.
- Both of the results tables are laid out well to allow for comparison between different models and the contribution of different components in the authors models from an ablation study.
- A comprehensive analysis of the models implemented, with respect to both baseline models and different constraint types.
- The ablation study gives a very good description of how the different components impact the performance of the model.

**Subreviewer:**

I delegated this review to a subreviewer.

**Weak Points:**

- Some choices are made without justification such as the Adam Optimiser and the data set splitting strategy.
- Could the “constraint shape” strategy potentially lose some useful relationships even though it is likely to help the model fit better for the majority of samples?  It wasn’t clear if the authors tested a model with both constraint ID embeddings and constraint shape embeddings.
- Minor: In the first paragraph of the Introduction the authors write the following without a reference:
For example, Wikidata has 1M “domain” constraint violations and 4.4M “single value” constraint violations as of March 20th, 2020.
Minor - I personally don’t like the heavy use of mathematical notation inside of textual paragraphs, particularly in the “Preliminaries” and the start of the “Baseline Approach” section.  It is less of an issue in other sections as it is used more sparingly or broken into lists.  I assume that this is to help with keeping the page count down but it makes the content more difficult to consume.  However, I find this issue with most of the computer science papers that I read.

---

> ### Author Rebuttal · Authors · 2021-01-28
>
> Thank you so much for your thoughtful review.
>
> We have chosen the Adam optimizer because it's quite standard in the neural network community and seemed to work well with a not too long training time. Investigating other optimizers is interesting indeed.
> We have also tested a model with both constraint ID embeddings and constraint shape embeddings, but it did not yield better results. We have therefore omitted it in the interest of space.
>
> Thank you again!

---

### Decision · Program_Chairs · 2021-02-23

**Decision:**

Accept

**Comment:**

This is a relevant work for SW that focuses on repairing RDF knowledge bases using neural networks over edit histories to learn to automatically correct constraint violations. It was applied to Wikidata and achieved improved results over previous work. The paper is well-written in general but could benefit from some clarifications.


**Strong points:**
- relevant topic and improved results
- extensive evaluation and new dataset
- open data and code

**Weak points:**
- motivation could be improved
- clarification of technical aspects and choices could be improved